# TaCA: Hot-Plugging Upgrades for Foundation Model with Task-agnostic Compatible Adapter

## Abstract

Visual foundation models, such as CLIP, exhibit exceptional proficiency in learning feature representations from extensive datasets via self-supervised techniques, showcasing noteworthy aptitude for transfer learning and generalization. A growing number of applications based on visual foundation models are emerging, including innovative solutions such as BLIP-2. These applications employ pre-trained CLIP models as upstream feature extractors and train various downstream modules to accomplish diverse tasks. However, scenarios necessitating system upgrades that entail updating the foundational model pose challenges, as they entail the inefficient and inflexible process of retraining all downstream modules to align with the new foundational model. In this paper, we propose an innovative and valuable task, *Hot-Plugging Upgrades for visual foundation models*. The aim is to seamlessly integrate superior-performing foundation models into downstream applications without adjusting the downstream modules. To realize this objective, we introduce a parameter-efficient and *T*ask-*a*gnostic *C*ompatible *A*dapter, referred to as TaCA, which promotes compatibility across distinct foundation models while concurrently enhancing performance for the new models. We conduct extensive experimental validation of TaCA using different scales of models with up to one billion parameters on various tasks such as video-text retrieval, video recognition, and visual question answering. The results consistently affirm the efficacy of TaCA in facilitating hot-plugging upgrades for visual foundation models. Codes and models will be made available.

## 1 Introduction

The data-centric methods of deep learning have catalyzed a massive increase in dataset scales and model sizes. Consequently, the exploration of versatile large models pre-trained (Radford et al., 2021; Jia et al., 2021; Caron et al., 2021; Oquab et al., 2023; Dosovitskiy et al., 2020) for various downstream tasks (Ju et al., 2022; Ni et al., 2022; Wang et al., 2021) is becoming a standard paradigm due to its enhanced performance and rapid convergence.

In light of this, a series of applications based on large-scale visual foundation models (*e.g.*, the dominant CLIP (Radford et al., 2021)) are burgeoning, which freeze the foundation models and train diverse modules to exploit pre-trained representations for downstream tasks. For example, to tackle the task of video classification, FrozenCLIP (Lin et al., 2022) equipped the pre-trained CLIP model with a lightweight Transformer decoder for spatiotemporal reasoning. The CLIP model has also been widely used as a visual tokenizer for large language models (LLMs) to form a multimodal LLM, *e.g.*, BLIP-2 (Li et al., 2023).

It poses a great challenge for upgrading the foundation models (*e.g.*, replacing the CLIP-ViT-B/16 with CLIP-ViT-L/14) accompanied by plenty of applications. A straightforward solution is to re-train all downstream modules, such as adapters or decoders, to adapt the new foundation models. However, task-oriented adaptation incurs substantial training costs, especially as the number of downstream tasks increases, rendering an impractical and inefficient solution.

In this paper, we break new ground by introducing the task of **hot-plugging upgrades for visual foundation models** in a modular framework, where the new foundation models can be directly deployed without any downstream adaptation. The representations encoded by the new foundation model are regularized to be compatible with the original old ones in the latent space. The compatible

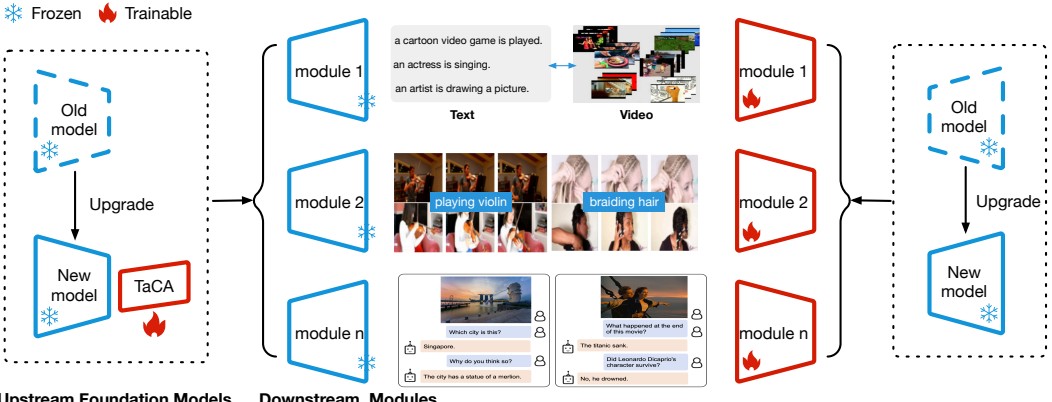

Figure 1: Different upgrading paradigms for visual foundation models. **(a)** Our approach enables hot-plugging upgrades of new foundation models while keeping the downstream modules untouched. Specifically, we train a task-agnostic compatible adapter (TaCA) on top of the new model, ensuring compatibility with both the old foundation models and the existing downstream modules. **(b)** In contrast, the paradigm of cold-plugging upgrades requires retraining the downstream modules whenever a new foundation model is introduced. This approach can be inefficient, particularly when dealing with a growing number of downstream applications.

new foundation model can be, therefore, directly integrated into the framework to work together with the pre-learned downstream modules, enabling a seamless upgrade of the whole framework and harvesting the benefits of upgraded foundation models immediately.

There are two main objectives for pursuing a compatible new foundation model. (i) The overall performance of the framework is anticipated to be enhanced upon the upgrade of the foundation model. In the most unfavorable scenario, where the new foundation model becomes indistinguishable from its predecessor, it maintains full compatibility with downstream modules; however, in such a circumstance, the upgrade would prove inconsequential for the system. (ii) The new foundation model can reliably enhance known or novel tasks using pre-learned adaptation modules from the old model. This encourages a task-agnostic learning approach, advocating for compatibility learning in new foundation models independent of any downstream task.

To this end, we propose a **T**ask-**a**gnostic **C**ompatible **A**dapter (**TaCA**) for the pre-trained, frozen new foundation model to attain compatibility with the original model. TaCA concurrently aligns new and old representation spaces while retaining the new model's strong capabilities through parameter-efficient tuning. Featuring lightweight adapters in each block of the new model and a dimension-alignment projector, TaCA is optimized toward alignment objectives, derived solely from the old foundation model, without requiring assistance from any downstream modules. TaCA takes the first step toward hot-plugging upgrades for foundation models, a practical task in the era of large models. We hope that our work would inspire future study and ease the upgrades of the foundation models in real-world applications.

We verify our approach using four different scales of visual foundation models with up to one billion parameters, *i.e.*, ViT-B/16, ViT-L/14, ViT-H/14, and ViT-G/14, all of which are pre-trained by the CLIP strategy. To illustrate the extent of TaCA's generalization, we conduct experiments on different backbones including CLIP (Radford et al., 2021), MAE (He et al., 2022b), Swin Transformer (Liu et al., 2022), DINO (Oquab et al., 2023), and BEiT (Bao et al., 2021). As for the downstream benchmarks, we adopt the modular frameworks, CLIP4Clip (Luo et al., 2022), FrozenClip (Lin et al., 2022), OpenFlamingo (Alayrac et al., 2022) and BLIP-2 (Li et al., 2023), to evaluate video-text retrieval, video classification, and visual question answering. For instance, by replacing the original ViT-B/16 with TaCA-ViT-H/14, performance improvements are observed on all ranges of benchmarks, *i.e.*, +4.9% on MSR-VTT (Xu et al., 2016) retrieval, +2.1% on Kinetics-400 (Kay et al., 2017) classification. Our model (TaCA-ViT-H/14) achieves +0.5% improvement compared with OpenFlamingo benchmark (ViT-L/14) on VQAv2 (Goyal et al., 2017). As shown in Figure

1, foundation models trained with TaCA can be hot-plugged into various modular frameworks for consistent improvements, and TaCA is proven to be effective for different model sizes.

To summarise, our main contributions are three-fold.

- We spearhead the exploration into the scenario of upgrading large-scale foundation models, and introduce a new task, *i.e.*, hot-plugging upgrades of visual foundation models in modular frameworks, which aims to harvest the benefit from new foundation models without necessitating the retraining process of downstream adaptation modules.

- To tackle the challenge, we introduce a parameter-efficient upgrading strategy using a Task-agnostic Compatible Adapter (TaCA). TaCA enables the new-old compatibility between foundation models and facilitates downstream applications in seamlessly integrating the new foundation model.

- Our method is comprehensively evaluated across a diverse range of downstream tasks, including video-text retrieval, video classification, and visual question answering. The results, displaying marked improvement, endorse the effectiveness and generalization ability of TaCA.

## 2 RELATED WORKS

**Visual foundation models.** In recent years, foundation models have witnessed a paradigm shift towards multi-modal supervision, demonstrating remarkable zero-shot performance across various downstream tasks. Numerous works (Ju et al., 2022; Ni et al., 2022; Wang et al., 2021; He et al., 2022c; Chen et al., 2021; d'Ascoli et al., 2021; Dong et al., 2022) harness large-scale image-text pairs culled from the Internet to learn visual representations and linguistic semantics via self-supervision. For instance, CLIP (Radford et al., 2021) constructed a text transformer and a visual transformer, which guides images and corresponding captions into a shared embedding space. Relying on the powerful generalization ability of foundation models, subsequent works explore the transfer of these pre-trained models to specific tasks. In this study, we concentrate on adapting CLIP for a range of downstream video tasks, including video-text retrieval (Luo et al., 2022; Jiang et al., 2022), video classification (Lin et al., 2022; Rasheed et al., 2022; Pan et al., 2022), and Video-QA (Li et al., 2023; Alayrac et al., 2022).

**Compatible representation learning** Backward Compatible Training (Shen et al., 2020; Zhang et al., 2021; Ramanujan et al., 2022; Zhang et al., 2022a; Su et al., 2022; Zhang et al., 2022b) has achieved notable success in the field of content-based image retrieval, which aims to make the features encoded by the new model interchangeable with those captured by the old model. Shen *et al* (Shen et al., 2020) devised the influence loss to inherit historical knowledge in the new model's training process, facilitating rapid system updates without backfilling gallery features. Zhang *et al* (Zhang et al., 2021) introduced the hot-refresh paradigm to incrementally enhance system performance via online refreshing. However, these methods encounter two main obstacles: (i) they require domain consistency in training datasets to maintain compatibility, meaning the new training set must be a superset of the old one, and (ii) they often require full fine-tuning of the backbone, which is unfeasible for large-scale foundation models.

**Parameter-efficient transfer learning** The emergence of large-scale pre-trained models has highlighted the need for efficient transfer of these foundation models to specific downstream tasks. Parameter-Efficient Transfer Learning (PETL) methods (Houlsby et al., 2019; Hu et al., 2021; Zhao et al., 2023; Lian et al., 2022; Chen et al., 2022b; Jie & Deng, 2022; Chen et al., 2022a) fix the original parameters and designs a few learnable parameters to overcome the domain gaps between the pretrained datasets and target tasks. The design of adapter (Houlsby et al., 2019) was first introduced in NLP, comprising a down-sampling layer and an up-sampling layer. A standard adapter is inserted into the Transformer (Vaswani et al., 2017) blocks (post self-attention operation and feed-forward operation). Hu *et al* (Hu et al., 2021) propose a method called LoRA, which maintains the pre-trained model weights and introduces trainable rank decomposition matrices into each layer of the Transformer architecture. Zhao *et al* (Zhao et al., 2023) bridge the domain gaps between the pre-trained and target datasets by aligning their distributions. Unlike existing adapter-based PETL methods where the adapters are task-specific, our objective is to achieve compatibility among various upstream foundation models. Consequently, our proposed compatible adapter is task-agnostic.

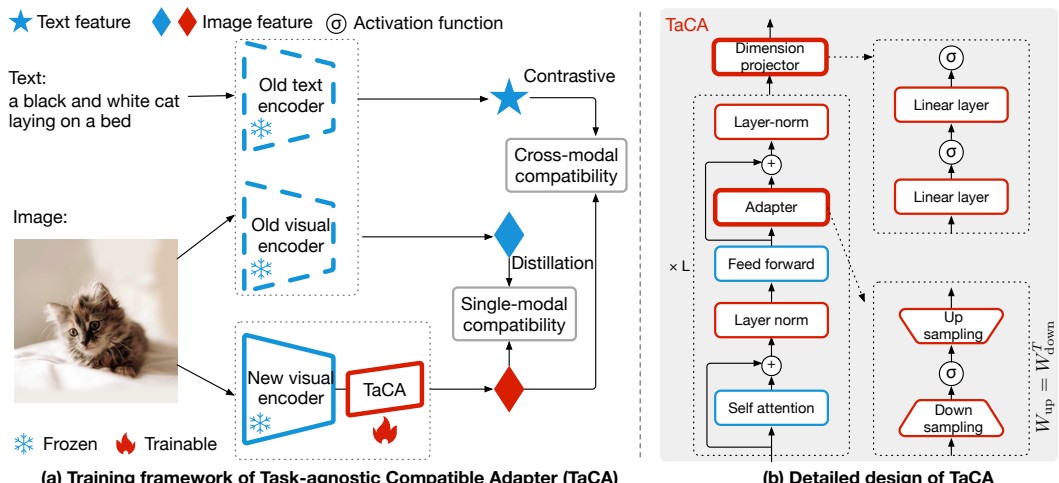

Figure 2: Our overall training framework, where the introduced task-agnostic compatible adapter (TaCA) is optimized to pursue compatibility between the new visual encoder and the old visual encoder. The training objectives of TaCA can be found in **(a)**, which consist of both single-modal compatibility (*i.e.*, image distillation loss in Eq. (5)) and cross-modal compatibility (*i.e.*, new-to-old contrastive loss in Eq. (6)). The detailed architecture of TaCA can be found in **(b)**, which consists of an adapter inserted into each Transformer block and a dimension projector.

## 3 METHODOLOGY

### 3.1 PRELIMINARIES

Our work aims to seamlessly upgrade the visual foundation models within a modular framework without the need to retrain the downstream modules. To illustrate this concept, we consider the widely used pre-trained CLIP models (Radford et al., 2021) as an example. CLIP models are extensively utilized as image feature extractors in various applications, including video-text retrieval, video classification, and visual question answering. By upgrading the visual foundation models, our approach facilitates enhanced performance and capabilities across these applications, without requiring extensive retraining or modifications to the downstream modules.

**Model architecture of CLIP.** CLIP adopts a dual-stream architecture, composed of a text Transformer (denoted as $\psi$) and a visual Transformer ($\phi$). CLIP, pre-trained by abundant image-text pairs, is capable of encoding the image and text semantics in a multimodal-aligned latent space. (i) Given an image $x_i^{\mathcal{I}} \in \mathcal{R}^{H \times W \times C}$ as input, it is first divided into $P \times P$ patches $x_i^{\mathcal{I}} \to \mathcal{R}^{HW/P^2 \times (P^2 C)}$, where $(H, W)$ denotes the resolution of the original image, and $C$ is the number of channels. The 2D patches are then flattened into a 1D sequence and fed into the visual Transformer, yielding the visual representation, *i.e.*, $\phi(x_i^{\mathcal{I}}) \in \mathbf{R}^d$. (ii) The corresponding descriptive text $x_i^{\mathcal{T}}$ is converted into embeddings using a tokenizer. Subsequently, these embeddings, supplemented with a `[CLS]` token at the beginning and a `[SEP]` token at the end of the sequence, are then processed through the text Transformer. We utilize `[SEP]` at the final layer as the text representation, *i.e.*, $\psi(x_i^{\mathcal{T}}) \in \mathbf{R}^d$.

**Training objective of CLIP.** CLIP is trained towards the alignment of image and text semantics using symmetric contrastive losses, which are formulated as NCE loss (He et al., 2020), such that

$$\mathcal{L}(\phi, \psi) = \frac{1}{|\mathcal{D}|} \sum_{x_i \in \mathcal{D}} \frac{1}{2} \left[ \text{NCE}(\phi(x_i^{\mathcal{I}}), \psi(x_i^{\mathcal{T}})) + \text{NCE}(\psi(x_i^{\mathcal{T}}), \phi(x_i^{\mathcal{I}})) \right],$$

$$s.t. \quad \text{NCE}(q, k) = -\log \frac{\exp(\langle q, k_+ \rangle / \tau)}{\sum_{i=1}^{\mathcal{B}} \exp(\langle q, k_i \rangle / \tau)}, \tag{1}$$

where $\phi$ is the visual encoder, $\psi$ is the text encoder, $\mathcal{B}$ is the mini-batch, $\mathcal{D}$ is the overall dataset, $\tau$ is the temperature hyper-parameter. $\langle \cdot, \cdot \rangle$ is the cosine similarity. The loss pulls closer positive image-text pairs while pushing away the mismatched pairs.

### 3.2 HOT-PLUGGING UPGRADES FOR VISUAL FOUNDATION MODELS IN MODULAR FRAMEWORKS

To tackle various downstream tasks, a commonly adopted approach is to utilize a foundation model that encodes generic and task-agnostic representations. These representations are then processed by downstream modules to achieve the final objective. In modular frameworks like this, upgrading the foundation model to improve upstream representations has involved the traditional *cold-plugging* paradigm. This paradigm requires retraining the downstream modules to ensure compatibility, which can be inflexible and costly, especially as the number of tasks increases.

We, therefore, introduce the task of *hot-plugging* upgrades for foundation models in visual tasks, where the new foundation model can be directly integrated into various modular frameworks with the pre-learned downstream modules untouched. To enable such a paradigm, the new foundation model should be compatible with the original one in terms of the encoded representations, that is, the new upstream features are interchangeable with the old ones.

**Task objectives.** There are two main objectives for pursuing a compatible new foundation model.

(i) *Performance gain:* As the baseline of model compatibility is to keep identical to the old model, where the downstream modules are naturally adapted, the most critical objective of hot-plugging upgrades is to achieve performance gains on the downstream task. Formally, we denote the evaluation metric for a specific task as $\mathcal{M}(\cdot)$ and the task head as $\zeta$. The objective can be formulated as

$$\mathcal{M}(\zeta_{\text{old}}(\phi_{\text{old}})) < \mathcal{M}(\zeta_{\text{old}}(\hat{\phi}_{\text{new}})) < \mathcal{M}(\zeta_{\text{new}}(\phi_{\text{new}})), \tag{2}$$

where $\mathcal{M}(\zeta_{\text{old}}(\phi_{\text{old}}))$ indicates the original performance of the system, $\mathcal{M}(\zeta_{\text{new}}(\phi_{\text{new}}))$ is the theoretical upper bound where the task head is specifically tuned for adapting the new foundation model (*i.e.*, the cold-plugging). Our compatible new foundation model $\hat{\phi}_{\text{new}}$ is expected to improve the overall system immediately without extra tuning of $\zeta_{\text{old}}$.

(ii) *Model versatility:* The above objective, *i.e.*, performance gain, can be achieved by using the old task head to produce learning targets for the new foundation model. However, we find that such task-oriented learning would hurt the generalization ability of the foundation models and lead to performance drops on novel tasks. A paradox, therefore, arises since the original intention of foundation models is to generalize across both known and unknown tasks. This encourages the new foundation models to learn compatibility in a task-agnostic manner, that is, the compatible new model $\hat{\phi}_{\text{new}}$ can be readily integrated into any framework and consistently improve the performance without accessing the downstream modules during training.

### 3.3 TaCA: TASK-AGNOSTIC COMPATIBLE ADAPTER

To tackle the above challenge, we introduce a Task-agnostic Compatible Adapter (TaCA) inserted in each Transformer block as shown in Figure 2. TaCA is trained on top of a pre-trained new visual foundation model, with the aim to project the new representations to the old latent spaces at the same time maintaining the stronger capability of the new model. For instance, we can improve the foundation model from ViT-B/16 to ViT-H/14 with TaCA.

**Architecture.** TaCA consists of a partly shared adapter (Houlsby et al., 2019) and a dimension projector for parameter-efficient tuning. The adapter is a bottleneck module with a down-sampling to reduce the feature dimension, a non-linear activation function, and an up-sampling to project back to the original dimension. As proved in prior research (Hu et al., 2021; He et al., 2022a), it has been established that parameter updates are predominantly situated within a low intrinsic dimension. In order to mitigate the training cost, we opt to employ **symmetric sampling matrices**. Given the input $x$ at the $i$-th layer, the output is formulated by,

$$\text{Adapter}(x) = x + W_{\text{up}} \cdot \sigma(W_{\text{down}} \cdot x), \tag{3}$$

$$W_{\text{up}} = (W_{\text{down}})^T, \tag{4}$$

where $W_{\text{down}} \in \mathbf{R}^{\mathbf{k} \times \mathbf{d}'}(d' \ll k)$ is the down-sampling matrix, $W_{\text{up}} \in \mathbf{R}^{\mathbf{d}' \times \mathbf{k}}$ is the up-sampling matrix, $\sigma$ is the activation function, $k$ is the input dimension, and $d'$ is the bottleneck dimension. The residual connection keeps the generalization ability of new models and helps the new models to fuse

the historical information of the old one. The dimension projector is formed with a two-layer MLP, which aligns the dimension of the new visual foundation model and the old one.

**Training objectives.** Intuitively, TaCA can be trained toward aligning the new and old representation spaces with a distillation loss. Formally,

$$\mathcal{L}_{\text{distill}}(\hat{\phi}_{\text{new}}; \phi_{\text{old}}) = \frac{1}{|\mathcal{D}|} \sum_{x_i^{\mathcal{I}} \in \mathcal{D}} \|(\hat{\phi}_{\text{new}}(x_i^{\mathcal{I}}), \phi_{\text{old}}(x_i^{\mathcal{I}})\|. \tag{5}$$

To further boost the compatibility of CLIP foundation models, we introduce the cross-model CLIP loss that maximizes the similarities of new visual features and matched old text features while minimizing the mismatched pairs, *i.e.*,

$$\mathcal{L}_{\text{contra}}(\hat{\phi}_{\text{new}}; \psi_{\text{old}}) = \frac{1}{|\mathcal{D}|} \sum_{x_i \in \mathcal{D}} \text{NCE}(\hat{\phi}_{\text{new}}(x_i^{\mathcal{I}}), \psi_{\text{old}}(x_i^{\mathcal{T}})). \tag{6}$$

The overall training objective can be formulated as

$$\mathcal{L}_{\text{TaCA}} = \mathcal{L}_{\text{contra}} + \lambda \mathcal{L}_{\text{distill}}, \tag{7}$$

where $\lambda$ is the loss weight and only the parameters of TaCA are optimized.

## 4 EXPERIMENTS

### 4.1 IMPLEMENTATION DETAILS

In this paper, we evaluate ViT-B/16, ViT-L/14, ViT-H/14, and ViT-G/14, all of which are pre-trained by CLIP and available from HuggingFace[1], ensuring ease of access and reproducibility. Specifically, we choose ViT-B/16 and ViT-L/14 as the old foundation models and test the performance gains when changing them to larger backbones. For the text encoders, we use the default 12-layer BERT model with modifications in GPT-2 (Radford et al., 2019).

We choose the well-known LAION-400M (Schuhmann et al., 2021) for TaCA training. In our experiments, the input image is resized into 224x224. The batch size is 64 for ViT-L/14 and 32 for both ViT-H/14 and ViT-G/14. The models are trained for 3 epochs using 8 NVIDIA A100 GPUs. We utilize the AdamW optimizer with a learning rate of $2e^{-4}$. The bottleneck dimension of the adapter is set to 128 for ViT-L/14 and 256 for both ViT-H/14 and ViT-G/14. $\lambda$ equals 2 in all of our experiments.

### 4.2 EFFICIENCY ANALYSIS

Each adapter layer of TaCA consists of $k \times d'$ learnable parameters, where $k$ is the input dimension and $d'$ is the bottleneck dimension of the adapter. The dimension projector consists of two-layer MLP with $d_n \times d_p + d_p \times d_o$, where $d_n, d_o$ are the output dimension of the new foundation model and the one of the old foundation model, $d_p$ is the hidden dimension (default as 4096). Thus the total trainable parameters are $Lkd' + d_n d_p + d_p d_o$, about 6% of the frozen parts, as shown in Table 11.

### 4.3 EXPERIMENTS ON VIDEO-TEXT RETRIEVAL WITH CLIP4CLIP

To assess the performance of TaCA on the video-text retrieval task, we employed CLIP4Clip (Luo et al., 2022) as our benchmark and conducted experiments on the MSR-VTT (Xu et al., 2016), MSVD (Chen & Dolan, 2011), and DiDeMo (Anne Hendricks et al., 2017) datasets. We only replace the old visual encoder with a new one while keep the other downstream modules and training details consistent with the original paper. The evaluation metrics used are text-to-video retrieval accuracy and video-to-text retrieval accuracy in terms of Recall@1.

For our evaluation, we selected ViT-B and ViT-L as the old visual foundation models to simulate two compatible scenarios. The results are presented in Table 1. As anticipated, our proposed method

---

[1]The checkpoints of ViT-B and ViT-L are downloaded from `https://huggingface.co/openai`, ViT-H and ViT-G are from `https://huggingface.co/models?library=open_clip`

Table 1: The results of video-text retrieval in terms of Recall@1. We use the CLIP4Clip (Luo et al., 2022) framework, which adopts CLIP-ViT models to extract frame features. We test our TaCA by upgrading the original visual foundation models (dubbed as VFM) with a larger one. We can observe that TaCA achieves consistent gains across diverse benchmarks with downstream modules untouched, indicating that the new VFM equipped with TaCA is powerful yet compatible. The de-emphasized lines report the theoretical upper bound for reference where the downstream modules/heads are retrained to adapt to the new incompatible VFM.

| Old VFM | New VFM | MSR-VTT | | MSVD | | DiDeMo | |
|---|---|---|---|---|---|---|---|
| | | T2V | V2T | T2V | V2T | T2V | V2T |
| ViT-B/16 | - | 42.1 | 40.2 | 45.2 | 58.9 | 36.2 | 34.0 |
| ViT-B/16 | TaCA-ViT-L/14 | 44.5 (+2.4) | 43.6 (+3.4) | 45.6 (+0.4) | 59.2 (+0.3) | 36.6 (+0.4) | 34.4 (+0.4) |
| - | ViT-L/14 | 44.8 | 44.5 | 46.7 | 60.1 | 38.1 | 36.4 |
| ViT-B/16 | TaCA-ViT-H/14 | 47.0 (+4.9) | 45.5 (+5.3) | 45.7 (+0.5) | 59.9 (+1.0) | 37.3 (+1.1) | 34.7 (+0.4) |
| - | ViT-H/14 | 47.5 | 46.4 | 48.2 | 61.7 | 39.0 | 37.2 |
| ViT-L/14 | - | 44.8 | 44.5 | 46.7 | 60.1 | 38.1 | 36.4 |
| ViT-L/14 | TaCA-ViT-H/14 | 47.2 (+2.4) | 46.0 (+1.5) | 47.1 (+0.4) | 60.5 (+0.4) | 38.4 (+0.3) | 37.0 (+0.6) |
| - | ViT-H/14 | 47.5 | 46.4 | 48.2 | 61.7 | 39.0 | 37.2 |
| ViT-L/14 | TaCA-ViT-G/14 | 47.8 (+3.0) | 46.3 (+1.8) | 47.3 (+0.6) | 61.0 (+0.9) | 39.1 (+1.0) | 37.4 (+1.0) |
| - | ViT-G/14 | 49.2 | 48.3 | 50.3 | 63.1 | 40.8 | 38.7 |

outperforms the baseline on all three datasets, demonstrating significant improvements. For instance, when replacing ViT-B with ViT-H, our method achieves a +4.9% increase on MSR-VTT, +0.5% on MSVD, and +1.1% on DiDeMo. However, the improvement difference between ViT-G → ViT-L and ViT-H → ViT-L is relatively minor, as the capability gap between the larger models narrows. Further studies on better large foundation models are called for.

## 4.4 EXPERIMENTS ON VIDEO CLASSIFICATION WITH FROZENCLIP

We evaluate our proposed TaCA on video classification tasks with the framework of FrozenClip (Lin et al., 2022), as illustrated in Table 2. The visual foundation model here is used for extracting frame features, which are then fed into the temporal reasoning module for classification.

Our observations are as follows: (i) Upgrading the visual foundation model leads to improved downstream performance. For instance, TaCA-ViT-L/14 achieves a 0.7% increase in accuracy on the Kinetics-400 dataset, and TaCA-ViT-H/14 demonstrates a significant improvement of 2.1%. (ii) The evaluated TaCA models are the same as those employed in the video-text retrieval task, meeting the requirements of hot-plugging upgrades (the compatible foundation model should improve various tasks consistently). The results indicate that our task-agnostic learning strategy is effective.

Table 2: The results of video classification in terms of top-1 accuracy. We utilize the FrozenClip (Lin et al., 2022) as the framework. The task heads are pre-learned with the old VFM, but can be directly used for the compatible new VFM equipped with TaCA.

| Old VFM | New VFM | K400 | UCF-101 |
|---|---|---|---|
| ViT-B/16 | - | 82.9 | 82.1 |
| ViT-B/16 | TaCA-ViT-L/14 | 83.6 (+0.7) | 83.1 (+1.0) |
| - | ViT-L/14 | 87.0 | 85.7 |
| ViT-B/16 | TaCA-ViT-H/14 | 85.0 (+2.1) | 84.2 (+2.1) |
| - | ViT-H-14 | 89.2 | 87.3 |
| ViT-L/14 | - | 87.0 | 85.7 |
| ViT-L/14 | TaCA-ViT-H/14 | 87.3 (+0.3) | 86.1 (+0.4) |
| - | ViT-H/14 | 89.2 | 87.3 |
| ViT-L/14 | TaCA-ViT-G/14 | 87.5 (+0.5) | 86.3 (+0.6) |
| - | ViT-G/14 | 90.5 | 88.4 |

## 4.5 EXPERIMENTS ON VISUAL QUESTION ANSWERING WITH MULTIMODAL LLM

To evaluate the performance of TaCA on the visual question answering, we use BLIP-2 (Li et al., 2023) and OpenFlamingo (Alayrac et al., 2022) as our benchmark methods and evaluate on the VQAv2 dataset. The evaluation metric used is accuracy. The results, as shown in Table 3, indicate that the new foundation model equipped with TaCA achieves a +0.6% and +0.4% improvement compared

Table 3: The results on visual question answering by the recent multimodal LLMs in terms of accuracy. The new VFM trained by TaCA can directly replace the original visual tokenizer without additional tuning for performance gains.

| Framework | VFM | VQAv2 (val) |
|---|---|---|
| OpenFlamingo | ViT-L/14 | 43.5 |
| | TaCA-ViT-H/14 | 44.0 (+0.5) |
| BLIP-2 | ViT-L/14 | 50.1 |
| | TaCA-ViT-H/14 | 50.5 (+0.4) |

Table 4: Ablation studies on different kinds of parameter-efficient tuning. We report the R@1 on MSR-VTT text-to-video retrieval and top-1 accuracy on K400.

| VFM | Type | MSR-VTT | K400 |
|---|---|---|---|
| ViT-B/16 (old) | - | 42.1 | 82.9 |
| TaCA-ViT-L/14 | LoRA | 44.0 | 80.3 |
| TaCA-ViT-L/14 | Adapter (std) | 44.7 | 83.4 |
| TaCA-ViT-L/14 | Adapter (ours) | 44.5 | 83.6 |

Table 5: Ablation studies on the bottleneck dimension of TaCA architecture. We report the R@1 on MSR-VTT text-to-video retrieval.

| VFM | $d'$ | MSR-VTT |
|---|---|---|
| ViT-B/16 (old) | - | 42.1 |
| TaCA-ViT-L/14 | 64 | 43.3 |
| TaCA-ViT-L/14 | 128 | 44.5 |
| TaCA-ViT-L/14 | 256 | 45.1 |
| TaCA-ViT-H/14 | 128 | 45.9 |
| TaCA-ViT-H/14 | 256 | 47.2 |
| TaCA-ViT-H/14 | 512 | 47.4 |

Table 6: Ablation studies on the training objectives. We report the R@1 on MSR-VTT text-to-video retrieval and top-1 accuracy on K400.

| VFM | MSR-VTT | K400 |
|---|---|---|
| ViT-B/16 (old) | 42.1 | 82.9 |
| TaCA-ViT-L/14 (w/o $\mathcal{L}_{contra}$) | 42.3 | 82.7 |
| TaCA-ViT-L/14 ($\lambda = 0$, w/o $\mathcal{L}_{distill}$) | 42.9 | 83.0 |
| TaCA-ViT-L/14 ($\lambda = 1$) | 44.1 | 83.3 |
| TaCA-ViT-L/14 ($\lambda = 2$) | 44.5 | 83.6 |
| TaCA-ViT-L/14 ($\lambda = 5$) | 44.4 | 83.5 |

to the original BLIP-2 model and the Flamingo model, respectively. These findings highlight the potential value of TaCA in easing the visual tokenizer upgrades in multimodal large language models.

## 4.6 ABLATION STUDIES OF TACA

**Effect of bottleneck dimensions.**    To analyze the impact of adapter size, we train different adapters with varying hidden dimensions. The results are presented in Table 5, revealing a consistent performance improvement as the dimension increases. While the performance gain for TaCA-ViT-L/14 is not substantial when increasing $d'$ from 128 to 256, we opt to set the default adapter dimension as 128 for the sake of computational efficiency. TaCA-ViT-H/14 exhibits a similar trend, and we choose 256 for a trade-off between efficiency and accuracy.

**Different kinds of parameter-efficient tuning.**    We conducted a comparison between three methods in parameter-efficient transfer learning, LoRA (Hu et al., 2021), standard Adapter (std) (Houlsby et al., 2019) and our proposed symmetric Adapter (ours). The results are summarized in Table 4. LoRA achieves performance gains on the MSR-VTT but experiences a degradation on K400, suggesting a limited generalization ability. In contrast, the Adapter architecture demonstrates superior performance across a spectrum of tasks. Our symmetric adapter yields comparable results with the standard adapter, while concurrently reducing the number of trainable parameters by 40%. This highlights the effectiveness of TaCA in achieving superior results and maintaining generalization capabilities. For additional results, please refer to the appendix.

**Discuss training objectives.**    To demonstrate the necessity of the proposed compatible losses, we conducted ablation studies, as shown in Table 6. We observed that using the single-modal distillation loss alone fails on video classification with performance degradation. On the other hand, while using a sole cross-model contrastive loss can enhance performance on both downstream tasks, the full model achieves optimal performance. The contrastive loss serves as an assurance of both the discriminativeness and compatibility of the new model, while the distillation loss functions to enhance its compatibility further. Furthermore, we investigated the effect of different loss weights ($\lambda$) by varying it from 1 to 5. We choose 2 empirically for optimal performance across tasks. These findings highlight the synergistic effect of the single-modal compatible loss and cross-modal compatible loss.

Table 7: Results on MSR-VTT dataset when varying training sets for TaCA. We report R@1 text-to-video retrieval.

| VFM | Training Set | T2V | V2T |
|---|---|---|---|
| ViT-B/16 (old) | - | 42.1 | 40.2 |
| TaCA-ViT-L/14 | CC12M | 43.1 | 42.1 |
| TaCA-ViT-L/14 | LAION-400M | 44.5 | 43.6 |
| TaCA-ViT-H/14 | CC12M | 45.5 | 44.3 |
| TaCA-ViT-H/14 | LAION-400M | 47.2 | 46.0 |

Table 8: Study the task-oriented fine-tuning of TaCA, which leads to in-domain gains while sacrificing generalization ability. We report results on MSR-VTT and K400.

| VFM | Finetune on | MSR-VTT | K400 |
|---|---|---|---|
| ViT-B/16 (old) | - | 42.1 | 82.9 |
| TaCA-ViT-L/14 | - | 44.5 | 83.6 |
| TaCA-ViT-L/14 | MSR-VTT | 45.0 ↑ | 80.4 ↓ |
| TaCA-ViT-L/14 | K400 | 41.2 ↓ | 86.8 ↑ |

Table 9: Generalization analysis under different backbones on video-text retrieval and video classification tasks. We report the R@1 on MSR-VTT and top-1 accurat on K400.

| VFM | Pretrain data | MSR-VTT | | K400 |
|---|---|---|---|---|
| | | T2V | V2T | Acc. |
| TaCA-CLIP-ViT-B/16 (old) | WiT | 42.1 | 40.2 | 82.9 |
| TaCA-MAE-ViT-L/14 | ImageNet-1k | 38.5 | 38.2 | 82.6 |
| TaCA-SwinV2-L/14 | ImageNet-22k | 37.9 | 37.5 | 82.7 |
| TaCA-DinoV2-ViT-L/14 | LVD | 42.6 (+0.5) | 41.3 (+1.1) | 83.2 (+0.3) |
| TaCA-BEiT-ViT-L/14 | ImageNet+DALLE | 42.3 (+0.3) | 40.6 (+0.4) | 83.1 (+0.2) |
| TaCA-CLIP-ViT-L/14 | WiT | 44.5 (+2.4) | 43.6 (+3.4) | 83.6 (+0.7) |

**Different training sets.** The image diversity of the training set plays a crucial role in determining the generalization ability of TaCA models. To investigate the impact of different training sets, we compare two popular pretraining datasets: LAION-400M (Schuhmann et al., 2021) and CC12M (Changpinyo et al., 2021). The results in Table 7 demonstrate that the larger LAION-400M can significantly enhance performance. The observation highlights the importance of a diverse and comprehensive training set in achieving improved compatibility and better generalization capabilities.

**Trade-off between generalization and specialization.** To demonstrate the effectiveness of our introduced task-agnostic training in terms of both performance gains on downstream tasks and generalization ability across different tasks, we conducted further fine-tuning on the Model (TaCA-ViT-L) using the MSR-VTT (or Kinetics-400) dataset. The results, as shown in Table 8, indicate that finetuning indeed improves in-domain performance. However, it harms the generalization ability, achieving much inferior performance on out-of-the-domain datasets.

**Different backbones.** We assess the generalization of TaCA across a spectrum of foundational backbones, as detailed in Table 9. Our observations are as follows: (1) When employing the same architecture, 'TaCA-CLIP-ViT-L/14' achieves the most favorable performance outcome. (2) With varying architectural configurations, DinoV2 and BEiT exhibit marginal enhancements but manage to meet the compatibility criterion across both tasks. This suggests that augmenting the scale of pretraining data can alleviate limitations when transitioning to downstream tasks. (3) In contrast, MAE and SwinV2 demonstrate suboptimal performance. Their underwhelming results may be attributed to weaker self-discrimination and transferability capabilities compared to the old CLIP model. This highlights that downstream tasks experience improvements only when the new foundational models exhibit enhanced representational capabilities.

## 5 CONCLUSIONS AND DISCUSSIONS

Visual foundation models have demonstrated powerful capabilities and are widely used as upstream feature extractors for various applications. To improve overall system performance and user experience, it is crucial to upgrade these foundation models over time. In this paper, we present a novel task called hot-plugging upgrades for visual foundation models within a modular framework. This framework allows for the direct deployment of new foundation models without any downstream adaptation. To achieve this, we propose a Task-Agnostic Compatible Adapter (TaCA), which ensures compatibility between the new and original models and preserves the generalization ability of the new model through parameter-efficient tuning. Extensive experiments demonstrate that TaCA maintains model versatility across various downstream tasks. Overall, our work introduces a pioneering approach to address the important task of upgrading foundation models, which has significant implications for the development and deployment of large-scale models in real-world applications.

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

APPENDIX

## A    EXPERIMENTAL DETAILS

### A.1    ALLOCATION OF TRANSFORMER-BASED VISUAL BACKBONES

To illustrate the extent of TaCA's generalization, we initially allocate the most widely used ViT-based foundational models that were pretrained on various datasets. This allocation is detailed in Table 10.

Table 10: Common transformer-based visual backbones.

| VFM | Pretrain data source | Data type | Data size | Available |
|---|---|---|---|---|
| CLIP-ViT-B (Radford et al., 2021) | WiT | Image-Text | 400M | x |
| CLIP-ViT-L (Radford et al., 2021) | WiT | Image-Text | 400M | x |
| OpenCLIP-ViT-L[2] | LAION | Image-Text | 400M | ✓ |
| MAE-ViT-L (He et al., 2022b) | ImageNet-1k | Image | 1M | ✓ |
| SwinV2-L (Liu et al., 2022) | ImageNet-22k | Image | 14M | ✓ |
| DinoV2-ViT-L (Oquab et al., 2023) | LVD | Image | 142M | x |
| BEiT-ViT-L (Bao et al., 2021) | ImageNet+DALLE | Image | 250M | x |

### A.2    EFFICENCY AND EFFECTIVENESS ANALYSIS

As shown in Table 11, our proposed symmetric adapter requires only 6% trainable parameters of the frozen parts while the standard one needs about 10%.

Table 11: Tunable parameters for TaCA. "VFM" is short for visual foundation model.

| VFM | Layers | Frozen params. | Learnable params. (our adapter) | Learnable params. (standard adapter) |
|---|---|---|---|---|
| ViT-B/16 | 12 | 86M | - | - |
| TaCA-ViT-L/14 | 24 | 307M | 17M | 37M |
| TaCA-ViT-H/14 | 32 | 632M | 42M | 70M |
| TaCA-ViT-G/14 | 40 | 1011M | 64M | 111M |

Additionally, we conduct a comparative analysis on video-text retrieval and video classification tasks. The outcomes presented in Table 12 affirm that our proposed adapter consistently attains performance levels comparable to the standard adapter, with only marginal sacrifice.

Table 12: The results of video-text retrieval in terms of Recall@1.

| Adapter | VFM | MSR-VTT | | MSVD | | DiDeMo | |
|---|---|---|---|---|---|---|---|
| | | T2V | V2T | T2V | V2T | T2V | V2T |
| - | ViT-B/16 (old) | 42.1 | 40.2 | 45.2 | 58.9 | 36.2 | 34.0 |
| Standard | TaCA-ViT-L/14 | 44.7 (+2.6) | 43.7 (+3.5) | 45.5 (+0.3) | 59.1 (+0.2) | 36.8 (+0.6) | 34.5 (+0.5) |
| Ours | TaCA-ViT-L/14 | 44.5 (+2.4) | 43.6 (+3.4) | 45.6 (+0.4) | 59.2 (+0.3) | 36.6 (+0.4) | 34.4 (+0.4) |
| Standard | TaCA-ViT-H/14 | 47.2 (+5.1) | 45.7 (+5.5) | 45.9 (+0.7) | 59.7 (+0.8) | 37.3 (+1.1) | 34.8 (+0.5) |
| Ours | TaCA-ViT-H/14 | 47.0 (+4.9) | 45.5 (+5.3) | 45.7 (+0.5) | 59.9 (+1.0) | 37.3 (+1.1) | 34.7 (+0.4) |

## B    ADDITIONAL ABLATION STUDY

**Inserted place of TaCA.**    Additionally, we investigate the impact of the insertion points of TaCA within the model architecture. We compare three different training settings, as presented in Table 13, where TaCA is inserted into different layers of the model. The results demonstrate that inserting

adapters into the shadow layers leads to better performance compared to inserting them into the deep layers (as indicated by line 1 and line 2 in the table). Moreover, when adapters are inserted into all layers, the model achieves state-of-the-art performance. This suggests that larger-scale foundation models require more learnable space to effectively transfer knowledge and enhance performance.

Table 13: Ablation studies on inserted place of TaCA architecture. We report the R@1 on MSR-VTT text-to-video retrieval.

| VFM | Inserted layers | MSR-VTT |
|---|---|---|
| ViT-B/16 (old) | - | 42.1 |
| TaCA-ViT-L/14 | 1-12 | 43.5 (+1.4) |
| TaCA-ViT-L/14 | 13-24 | 42.6 (+0.5) |
| TaCA-ViT-L/14 | 1-24(full) | 44.5 (+2.4) |

**Effect of pre-train datasets.** To prove that our method is not constrained with the same pre-train dataset, we conducted additional experiments, the results of which are presented in Table 14. Due to unaligned dimensions, we add a learnable linear layer after the output layer of the new model as baselines (denoted as '+Linear').

Table 14: Ablation studies on pre-train datasets. We report the R@1 on MSR-VTT dataset.

| VFM | Pretrain data | MSR-VTT | |
|---|---|---|---|
| | | T2V | V2T |
| CLIP-ViT-B (old) | WiT | 42.1 | 40.2 |
| CLIP-ViT-L (+Linear) | WiT | 35.4 | 34.7 |
| TaCA-CLIP-ViT-L | WiT | 44.5 (+2.4) | 43.6 (+3.4) |
| OpenCLIP-ViT-L (+Linear) | LAION | 31.5 | 31.6 |
| TaCA-OpenCLIP-ViT-L | LAION | 44.0 (+1.9) | 43.1(+2.9) |

Our observations are as follows: (1) A comparison between "CLIP-ViT-L (+Linear)" and "OpenCLIP-ViT-L (+Linear)" highlights that similar pre-training data can offer better initialization, contributing to the alignment of feature spaces. (2) Despite the "OpenCLIP-ViT-L (+TaCA)" model being initialized from a distinct pre-training dataset, it still exhibits notable enhancements in comparison to the linear model. Notably, the model with the same initialization ("CLIP-ViT-L (+TaCA)") performs optimally. (3) Our TaCA method produces improvements, whereas the baseline (Linear) fails to meet compatibility requirements.

## C    INFERENCE DETAILS

The objective of hot-plugging model upgrades for visual foundation models is to seamlessly integrate the new foundation model into different modular frameworks using a Task-Agnostic Compatible Adapter (TaCA). This integration is achieved without modifying or retraining the pre-existing downstream modules. The training process of TaCA is designed to be independent of the specific downstream tasks, ensuring flexibility and efficiency in the upgrading process.

In the subsequent sections, we will provide detailed explanations of the upgrading process through TaCA in the domains of video-text retrieval, video classification, and visual question answering tasks. We will outline the specific steps and considerations involved in each task to ensure the successful integration and compatibility of the new foundation model within the respective frameworks.

## C.1 VIDEO-TEXT RETRIEVAL

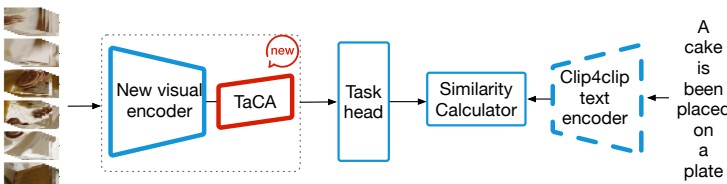

Figure 3: TaCA implementation in CLIP4Clip framework for video-text retrieval.

**Inference Framework of CLIP4Clip.** The implementation of hot-plugging model upgrades within the CLIP4Clip framework is depicted in Figure 3. In our implementation of the CLIP4Clip framework, we make several changes to adapt it for the upgrade scenario. We use the released codes from the provided repository [3], and incorporate the following modifications: Instead of finetuning all parameters in the visual and text backbones, we fix these backbones and focus on training different task heads. Specifically, we add a 4-layer MLP task head after the visual encoder. We trained these task heads using different downstream training datasets: WIT and MSR-VTT Training-7k for MSR-VTT, WIT and MSVD Training for MSVD, and WIT and DiDeMo Training for DiDeMo. These modifications allow us to adapt the CLIP4Clip framework for the upgrade scenario and conduct experiments accordingly.

**Test Dataset.** (i) **MSR-VTT** (Xu et al., 2016) is a dataset consisting of 10,000 videos and 200,000 captions. The test data subset contains 1,000 video-text pairs for evaluation purposes. (ii) **MSVD** (Chen & Dolan, 2011) comprises 1,970 videos, which is divided into train, validation, and test splits, containing 1,200, 100, and 670 videos, respectively. Each video is associated with approximately 40 sentences. (iii) **DiDeMo** (Anne Hendricks et al., 2017) consists of 10,000 videos, each of which is annotated with four sentences, resulting in a total of 40,000 sentences. In the evaluation of video-paragraph retrieval, following the approach proposed by Liu *et al.* (Liu et al., 2019), all the sentence descriptions for a particular video are concatenated into a single query.

**Metric.** In evaluating the performance of our model, we utilize the standard retrieval metric known as recall at rank K (R@K), where a higher value indicates better performance. R@K measures the percentage of test samples for which the correct result is found within the top-K retrieved items. More specifically, we calculate R@K for both text-to-video and video-to-text retrieval. For text-to-video retrieval, we measure the retrieval accuracy when the query is a text input. Conversely, for video-to-text retrieval, we assess the retrieval accuracy when the query is a video input.

**Inference Details.** To ensure consistency with other downstream tasks, we utilize CLIP-ViT-B/16 as the pretrained model, whereas the original paper employ CLIP-ViT-B/32. In aggregating the features of all frames, we directly use "mean pooling" to obtain an "average frame". During inference, only the old visual encoder is re-deployed with the new one, while the other downstream modules, including the text encoder and similarity calculator, remain fixed and unchanged. The implementation allows for the seamless integration of upgraded visual encoders into the existing CLIP4Clip framework, enabling improved performance and capabilities in video-text retrieval tasks without the need to modify or retrain the text encoder and similarity calculator components.

## C.2 VIDEO CLASSIFICATION

**Inference Framework of FrozenClip.** The implementation of hot-plugging model upgrades for FrozenClip (Lin et al., 2022) framework is shown in Figure 4.

**Test Dataset.** (i) **Kinetics-400** dataset comprises 240,436 training videos and 19,787 validation videos. It consists of 400 human action classes, with a minimum of 400 video clips available for each

---

[3]https://github.com/ArrowLuo/CLIP4Clip

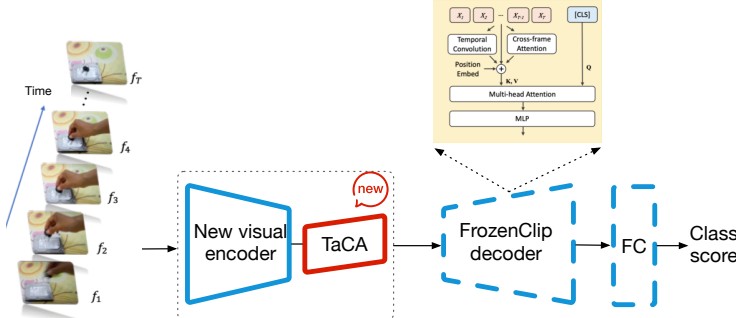

Figure 4: TaCA implementation in FrozenClip framework for video classification.

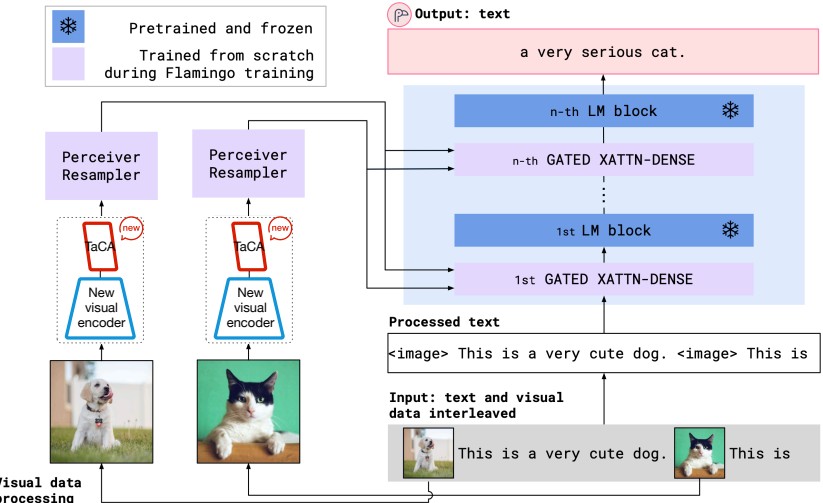

Figure 5: TaCA implementation in Flamingo framework for visual question answering.

action. (ii) **UCF-101** dataset consists of 13,320 video clips. These video clips are classified into 101 categories, representing various human actions and activities.

**Inference Details.** In our implementation, we maintain the same inference settings as described in the paper (Lin et al., 2022). We download all the pretrained modules from the provided repository, which can be accessed at the following link [4].

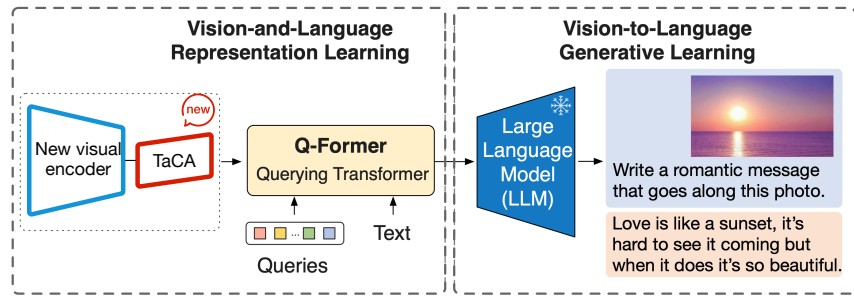

Figure 6: TaCA implementation in BLIP-2 framework for visual question answering.

## C.3 VISUAL QUESTION ANSWERING

**Inference Framework of Flamingo and BLIP-2.** The implementation of hot-plugging model upgrades for Flamingo (Alayrac et al., 2022) and BLIP-2 (Li et al., 2023) frameworks is shown in Figure 5 and Figure 6, respectively.

**Test Dataset.** VQA-v2 (Goyal et al., 2017) is a dataset specifically designed for visual question answering tasks. The dataset includes a total of 265,016 images, which encompass a combination of images from the COCO dataset and abstract scenes. For each image, there are a minimum of 3 questions, with an average of 5.4 questions per image. Additionally, there are 10 ground truth answers for each question.

**Inference Details.** In the Flamingo framework, we employ OpenFlamingo-9B (Clip-ViT-L/14) as the old visual encoder. As for the upgraded model, we utilize TaCA-ViT-H/14 as the new visual encoder. In the BLIP-2 framework, we choose BLIP-2-ViT-L-OPT$_{2.7B}$ as the old visual encoder. Similarly, the upgraded model incorporates TaCA-ViT-H/14 as the new visual encoder. These configurations allow us to compare and evaluate the performance of the new visual encoders (TaCA-ViT-H/14) against the existing models (OpenFlamingo-9B and BLIP-2-ViT-L-OPT$_{2.7B}$) within the Flamingo and BLIP-2 frameworks, respectively.

**Visualization.** We evaluate and compare the multi-modality capabilities of TaCA and Open-Flamingo using the open-ended VQAv2 dataset (Goyal et al., 2017). Figure 7 illustrates several positive examples, demonstrating our model's proficiency in processing complex semantics and extracting intricate visual information.

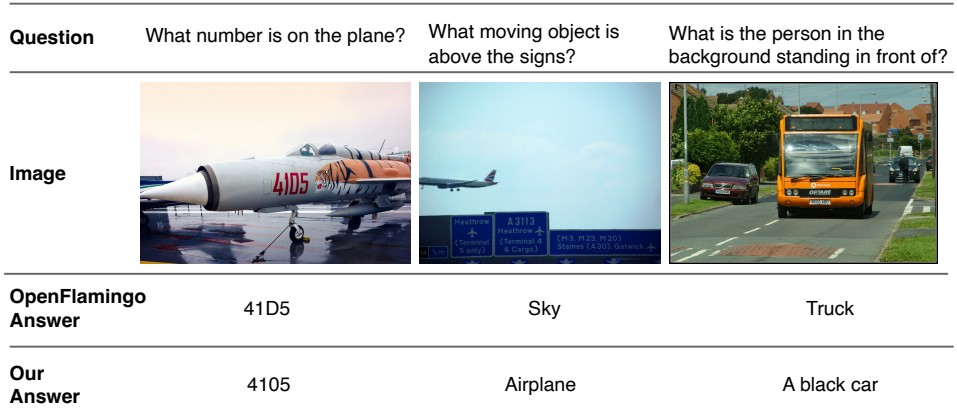

Figure 7: Visualization results under OpenFlamingo framework on VQAv2 dataset. The baseline method (Flamingo) utilizes ViT-L/14 as its foundation model. By replacing the old visual foundation model with TaCA-ViT-H/14, while keeping other downstream modules unchanged, our method shows notable enhancement in generating satisfactory answers.

## C.4 STABLE DIFFUSION

**Inference Framework of UNCLIP.** We have extended our investigation to encompass image generation as an additional downstream task. To achieve this, we harness the UNCLIP (Ramesh et al., 2022) framework, replacing the native image encoder (CLIP-ViT-L/14) with our novel encoder (TaCA-CLIP-ViT-G/14). The visualization is shown in Figure 8.

---

[4]https://github.com/OpenGVLab/efficient-video-recognition

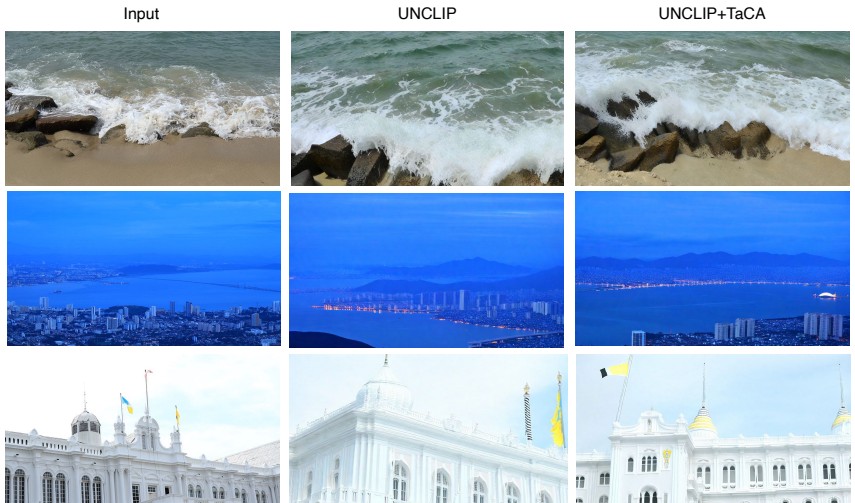

Figure 8: Image generation with Stable Diffusion. 'UNCLIP' utilizes CLIP-ViT-L as the image encoder, while 'UNCLIP+TaCA' replace the old encoder with 'TaCA-CLIP-ViT-H'.

## D  ADDITIONAL RELATED WORKS

**Knowledge Distillation.** The goal of knowledge distillation (KD) is to transfer the knowledge learned by the teacher model to the student model(Yang et al., 2022b;a). Typically, the teacher model is comprehensive while the student model is lightweight. Unlike the previous works, the hot-plugging upgrading desires the performance improvement via replacing the old visual foundation model with an advanced one.

## E  LIMITATIONS

It is worth mentioning that while TaCA achieves marginal improvements on certain downstream tasks, there is potential for further improvement through the use of more advanced adapter architectures or larger training sets. Additionally, the concept of compatible upgrades for foundation models should not be limited to the visual domain alone. Foundation models in other modalities, such as text encoders in Stable Diffusion (Rombach et al., 2022), also face similar challenges in terms of upgrades. We hope that our work serves as an inspiration for future research in this area and contributes to facilitating the upgrades of foundation models in real-world applications.

## F   PSEUDO CODE IN PYTORCH-STYLE

The pseudo code of hot-plugging model upgrades for visual foundation models is shown in Algorithm 1.

---

Algorithm 1: Pseudo code of Hot-Plugging Model Upgrades in a PyTorch-like style.

```
 1  # old_model: includes the pretrained visual encoder and the
        pretrained text encoder.
 2  # new_model: consists of fixed new visual encoder, trainable
        adapter and dimension projector.
 3  # tau: temperature, lambda: loss weight
 4
 5  # Freeze pretrained parameters in the backbones
 6  for param in [old_model.params(), new_model.params()]:
 7      param.requires_grad = False
 8
 9  # Set adapter and dimension projector be trainable
10  for param in new_model.params():
11      if 'adapter' in param or 'dim_projector' in param:
12          param.requires_grad = True
13
14  for (image, text) in loader: # load a mini-batch samples
15      with torch.no_grad():
16          old_visual_feat=old_model.visual_encoder.forward(image).
                detach()
17          old_text_feat=old_model.text_encoder.forward(text).
                detach()
18      new_visual_feat = new_model.visual_encoder.forward(image)
19      new_visual_feat = new_model.dim_projector.forward(
            new_visual_feat)
20
21      # Distillation loss, Eq.(4)
22      distill_loss = l2_loss(new_visual_feat, old_visual_feat)
23
24      # Contrastive loss, Eq.(5)
25      logits = bmm(new_visual_feat, old_text_feat.T) / exp(tau)
26      labels = arange(n)
27      loss_i = cross_entropy_loss(logits, labels, axis=0)
28      loss_t = cross_entropy_loss(logits, labels, axis=1)
29      contra_loss = (loss_i + loss_t)/2
30
31      # SGD update: adapter and dimension projector
32      loss = contra_loss + lambda*distill_loss
33      loss.backward()
34      update(new_model.params)
```

---

bmm: batch matrix-matrix product; .T: matrix transpose.

