# OpenReview forum: "TaCA: Hot-Plugging Upgrades for Foundation Model with Task-agnostic Compatible Adapter"
_ICLR.cc/2024/Conference — Submitted to ICLR 2024_

### Official Review · Reviewer_N62r · 2023-10-27

**Soundness:** 3 good
**Presentation:** 3 good
**Contribution:** 3 good
**Rating:** 6
**Confidence:** 5

**Summary:**

The authors propose a new task, Hot-Plugging Upgrades for visual foundation models. The aim is
to seamlessly integrate superior-performing foundation models into downstream applications without adjusting the downstream modules. To realize this objective, The authors introduce a parameter-efficient and Task-agnostic Compatible Adapter, referred to as TaCA, which promotes compatibility across distinct foundation models while concurrently enhancing performance for the new models. The authors conduct extensive experimental validation of TaCA using different scales of models with up to one billion parameters on various tasks such as video-text retrieval, video recognition, and visual question answering.

**Strengths:**

- The authors propose a hot-plugging upgrading module, which is interesting.
- The experiments have been conducted to illustrate the superiority of the proposed method.

**Weaknesses:**

- The authors should validate the flexibility of the proposed TaCA module. How about the performance when the TaCA is aligned with the other LLMs?
- The qualitative analysis and visualization in experiments are missing.

**Questions:**

- The authors should validate the flexibility of the proposed TaCA module. How about the performance when the TaCA is aligned with the other LLMs?
- The qualitative analysis and visualization in experiments are missing.

---

> ### Author Response · Authors · 2023-11-22
>
> **Q1: The authors should validate the flexibility of the proposed TaCA module. How about the performance when the TaCA is aligned with the other LLMs?**
>
> **A1:** The proposed TaCA demonstrates exceptional flexibility by seamlessly integrating with a range of downstream tasks, eliminating the need for additional fine-tuning costs. Its versatility has been validated across tasks such as video-text retrieval, video classification, visual question answering, zero-shot image classification, and zero-shot image-text retrieval. Table 3 further illustrates TaCA's compatibility with the BLIP-2 LLM, confirming its robust performance when paired with different language models.
>
>
>
> **Q2: Addressing the Lack of Qualitative Analysis and Visualizations**
>
> **A2:** We recognize the importance of qualitative analysis and visualization in our experimental assessment. The manuscript currently discusses qualitative aspects, and visualizations are provided through the inference framework and image generation samples in Appendix C. We will enhance our manuscript with a more comprehensive qualitative analysis and additional visualizations in line with your feedback.

---

### Official Review · Reviewer_REM4 · 2023-10-31

**Soundness:** 2 fair
**Presentation:** 3 good
**Contribution:** 2 fair
**Rating:** 5
**Confidence:** 3

**Summary:**

This paper introducing the task of hot-plugging upgrades for visual foundation models. And it proposes TaCA, which aims at effectively replacing visual foundation model without any downstream adaptation. Extensive experiments on video-related tasks indicate the effectiveness of TaCA.

**Strengths:**

1. This paper introduces the task of hot-plugging upgrades for visual foundation models, which aims at effectively replacing upstream visual fundation model.

2. The experimental results prove TaCA can upgrade the visual foundation models without requiring the training data from downstream video-related tasks.

**Weaknesses:**

TaCA forces large-scale visual model to align with relatively small-scale visual model using adapter. It defeats the purpose of changing the visual model in my opinion. This approach restricts the transferability of the large-scale visual model, which may limit its potential benefits. Additionally, according to the results presented in Table 2, TaCA provides marginal improvements on downstream video classification tasks comparing to directly using large-scale visual model.

This paper evaluates the effectiveness of TaCA on video-related tasks. However, there are a number of works transfer CLIP to image-based tasks, e.g. image-text retrieval, image segmetation, and few-shot image classification. The absence of experiments on image-related tasks in this paper leaves a gap in evaluating TaCA's capability.

**Questions:**

What is the training overhead in terms of time?

---

> ### Author Response · Authors · 2023-11-22
>
> **Q1: TaCA forces large-scale visual model to align with relatively small-scale visual model using adapter. in Table 2, TaCA provides marginal improvements on downstream video classification tasks comparing to directly using large-scale visual model.**
>
> **A1:** **The purpose of TaCA is to efficiently align the old model with a large-scale or powerful visual model, thereby enhancing downstream task performance.** In Table 2 in the main paper, the de-emphasized lines represent the results achieved by fine-tuning all downstream heads, which are task-specific and inefficient. Conversely, the emphasized lines depict the performance obtained by directly deploying the TaCA model without making any modifications to the downstream modules.
>
>
>
>
>
> **Q2: Add experiments on image-related tasks like image-text retrieval, image segmentation, and few-shot image classification.**
>
> **A2:** We have broadened the scope of our experiments to include fundamental image classification and retrieval tasks. The results, detailed in Tables 1 and 2, demonstrate TaCA's consistent performance improvements across these basic image tasks.
>
>
>
> | **Model**         | **ImageNet** | **CIFAR100** |
> | ----------------- | ------------ | ------------ |
> | **ViT-B/16**      | 68.6         | 68.7         |
> | **ViT-L/14**      | 75.3         | 77.9         |
> | **TaCA-ViT-L/14** | 74.3(+5.7)   | 76.5(+7.8)   |
>
> Table1: Zero-shot image classification on ImageNet and CIFAR100 datasets. We report the accuracy.
>
> | **Model**         | **MSCOCO**   | **Flickr**  |
> | ----------------- | ------------ | ----------- |
> | **ViT-B/16**      | 42.75        | 72.18       |
> | **ViT-L/14**      | 46.42        | 75.08       |
> | **TaCA-ViT-L/14** | 44.71(+3.67) | 73.97(+2.9) |
>
> Table2: Zero-shot image-text retrieval on MSCOCO and Flickr datasets. We report the Recall@1.
>
>
>
> **Q3: What is the training overhead in terms of time?**
>
> **A3:** The fine-tuning schedules and inference details for these paradigms are presented in Appendix C.  In the video-text retrieval task, we fine-tune three distinct sets of downstream heads on the MSRVTT, MSVD, and DiDeMo datasets, resulting in the training of a total of 42M parameters. For the video classification task, fine-tuning the classification head requires nearly twice as many parameters as the TaCA model. However, TaCA requires training just once, with no additional fine-tuning costs, yielding a more parameter-efficient and training-free application in downstream tasks. This discussion will be incorporated into the final paper.
>
>
>
> | **Strategy**   | **Learnable params.** | **Finetune data**      | **MSRVTT** | **MSVD** | **DiDeMo** |
> | -------------- | --------------------- | ---------------------- | ---------- | -------- | ---------- |
> | **Finetuning** | 14M                   | WIT+MSRVTT/MSVD/DiDeMo | 44.8       | 46.7     | 38.1       |
> | **TaCA**       | 17M                   | No need                | 44.5       | 45.6     | 36.6       |
>
> Table3: Comparison on Video-Text Retrieval datasets. The new visual fundation model (VFM) is ViT-L/14, and the old VFM is ViT-B/16.
>
> | **Strategy**   | **Learnable params.** | **Finetune data** | **K400** | **UCF-101** |
> | -------------- | --------------------- | ----------------- | -------- | ----------- |
> | **Finetuning** | 29M                   | Kinetics-400      | 87       | 85.7        |
> | **TaCA**       | 17M                   | No need           | 83.6     | 83.1        |
>
> Table4: Comparison on Video Classification datasets. The new visual fundation model (VFM) is ViT-L/14, and the old VFM is ViT-B/16.

---

> > ### Author Response · Authors · 2023-11-23
> > **Looking forward to the further discussion**
> >
> > Dear Reviewer #REM4,
> >
> > Thank you for your valuable time and constructive feedback. In response, we have revised the manuscript with the following updates:
> >
> > 1. We have refined our discussion of the paper's contributions. Our principal goal is the introduction of "hot-plugging upgrades" as a feasible strategy for incorporating new visual foundation models into a variety of downstream tasks. This method eliminates the necessity for resource-intensive retraining processes, clarifying any previous ambiguities related to weaknesses 1.
> >
> > 2. We have broadened the scope of our experiments to encompass additional basic image tasks. We now present results from zero-shot image classification and image-text retrieval, which address the issues highlighted in weaknesses 2 of your initial review.
> >
> >
> > We are eager to engage in further discussions and hope for your reconsideration of our work.

---

### Official Review · Reviewer_dvkW · 2023-10-31

**Soundness:** 2 fair
**Presentation:** 4 excellent
**Contribution:** 1 poor
**Rating:** 5
**Confidence:** 4

**Summary:**

This paper proposes Hot-Plugging Upgrades for visual foundation models. The aim is to seamlessly integrate superior-performing foundation models into downstream applications without adjusting the downstream modules. To realize this objective, this paper introduces a parameter-efficient and task-agnostic Compatible Adapter, referred to as TaCA, which promotes compatibility across distinct foundation models while concurrently enhancing performance for the new models. The paper is written well and easy to follow.

**Strengths:**

1. This paper spearheads the exploration into the scenario of upgrading large-scale foundation models and introduces hot-plugging upgrades of visual foundation models in modular frameworks.
2. This paper introduces a parameter-efficient upgrading strategy using a Task-agnostic Compatible Adapter (TaCA)
3. The paper is written well and easy to follow.

**Weaknesses:**

1. The approach is incremental, and the techniques employed are all verified strategies. Specifically, it utilizes a combination of distillation methods and contrastive learning, forming a hybrid approach.

2. Why not conduct experiments on more basic image classification and retrieval datasets (e.g., MSCOCO and imagenet)? If the effectiveness of this method can be verified on a more basic dataset, I am willing to increase my score

3. In my opinion, TaCA, which utilizes an adapter to align a large-scale visual model with a smaller-scale visual model, undermines the purpose of changing the visual model. This approach hampers the transferability of the large-scale visual model, potentially limiting its advantages. Moreover, based on the results presented in Table 2, TaCA only shows marginal enhancements in downstream video classification tasks compared to directly employing a large-scale visual model.

4. while this paper assesses the effectiveness of TaCA in video-related tasks, it overlooks numerous studies that apply CLIP to image-based tasks such as image-text retrieval, image segmentation, and few-shot image classification. The absence of experiments on image-related tasks in this paper creates a gap in evaluating TaCA.

5. What would happen if Old VFM and New VFM were different (e.g., VIT-B to ResNet-50)? Can we distill and transfer knowledge between VFMs of different architectures to each other? For example, distilling knowledge from miniGPT or LLAMA (decoder only architecture) to CLIP?

**Questions:**

1. The method is incremental, and the methods used are all validated schemes, which is actually distillation method combined with contrastive learning.
2. Why not conduct experiments on more basic image classification and retrieval datasets

---

> ### Author Response · Authors · 2023-11-22
>
> **Q1: The approach is incremental.**
>
> **A1:** We reclaim that our key contributions are:
>
> (1) For the **first time**, we introduce "hot-plugging upgrades", which directly deploys the new foundation models **without retraining cost**. Specifically, we introduce the **task-agnostic** compatible adapter (TaCA), which enhances performance across a variety of downstream tasks.
>
> (2) Our method not only demonstrates compelling practical value but also boasts remarkable industrial applications. Unlike the traditional approach of fine-tuning specific downstream heads, which often incurs prohibitively high training costs and exhibits suboptimal transferability, our approach shines. To illustrate, the process of retraining a Q-Former in BLIP-2 demands approximately 9 days utilizing 16 A100 GPUs. TaCA, in contrast, offers a versatile task-agnostic capability, enabling seamless adaptation to various downstream tasks without incurring any finetuning costs.
>
> (3) The significance and novelty of our contributions have been duly acknowledged by Reviewer #TD34 and #N62r, further attesting to the groundbreaking nature of our work.
>
>
>
>
>
> **Q2: Add experiments on more basic image classification and retrieval datasets (e.g., MSCOCO and imagenet)**
>
> **A2:** We have broadened the scope of our experiments to include fundamental image classification and retrieval tasks. The results, detailed in Tables 1 and 2, demonstrate TaCA's consistent performance improvements across these basic image tasks.
>
>
>
> | **Model**         | **ImageNet** | **CIFAR100** |
> | ----------------- | ------------ | ------------ |
> | **ViT-B/16**      | 68.6         | 68.7         |
> | **ViT-L/14**      | 75.3         | 77.9         |
> | **TaCA-ViT-L/14** | 74.3(+5.7)   | 76.5(+7.8)   |
>
> Table1: Zero-shot image classification on ImageNet and CIFAR100 datasets. We report the accuracy.
>
> | **Model**         | **MSCOCO**   | **Flickr**  |
> | ----------------- | ------------ | ----------- |
> | **ViT-B/16**      | 42.75        | 72.18       |
> | **ViT-L/14**      | 46.42        | 75.08       |
> | **TaCA-ViT-L/14** | 44.71(+3.67) | 73.97(+2.9) |
>
> Table2: Zero-shot image-text retrieval on MSCOCO and Flickr datasets. We report the Recall@1.
>
>
>
>
>
> **Q3: TaCA utilizes an adapter to align a large-scale visual model with a smaller-scale visual model. In Table 2, TaCA only shows marginal enhancements in downstream video classification tasks compared to directly employing a large-scale visual model**
>
> A3: We acknowledge the confusion in our initial explanation. **The purpose of TaCA is to efficiently align the old model with a large-scale or powerful visual model, thereby enhancing downstream task performance.** In Table 2, the de-emphasized lines represent the results achieved by fine-tuning all downstream heads, which are task-specific and inefficient. Conversely, the emphasized lines depict the performance obtained by directly deploying the TaCA model without making any modifications to the downstream modules.
>
>
>
> **Q4: Add experiments on image-related tasks like image-text retrieval, image segmentation, and few-shot image classification.**
>
> **A4:** For further details on the experiments encompassing image-text retrieval, image segmentation, and few-shot image classification, please refer to the response to Question 2.
>
>
>
> **Q5: What would happen if Old VFM and New VFM were different? Can knowledge of different architectures be distilled? For example, distilling knowledge from miniGPT or LLAMA (decoder only architecture) to CLIP?**
>
> **A5:** (1) Our research in Table 9 establishes that TaCA can generalize across various foundational models, allowing for knowledge distillation between different architectures. However, the performance may be less optimal if the new model is not as capable as the previous one (e.g., replacing ViT-B with ResNet-50). (2) While this paper confirms TaCA's efficiency with visual models, its applicability to language models, such as distilling knowledge from decoder-only architectures like miniGPT or LLAMA to CLIP, remains an area for future exploration.

---

> > ### Author Response · Authors · 2023-11-23
> > **Looking forward to the further discussion**
> >
> > Dear Reviewer #dvkW,
> >
> > Thanks for updating your comments, and we greatly value your time and insightful feedbacks. We have made amendments to the manuscript in light of your suggestions:
> >
> > 1. We have expanded our experimentation section to include additional basic image tasks. Specifically, we have conducted zero-shot image classification and zero-shot image-text retrieval experiments, addressing concerns raised in weaknesses 2 and 4.
> > 2. We have clarified the contribution of our study. Our primary aim is to introduce the concept of "hot-plugging upgrades." This is a practical methodology for the integration of new visual foundation models into existing downstream tasks without the need for resource-intensive retraining, thereby resolving misunderstandings mentioned in weakness 1 and 3.
> >
> > We believe that the value of a paper should not be judged solely on the novelty of methods proposed, but also on the introduction of substantial and previously unexplored tasks. We look forward to your reconsideration and welcome further dialogue.

---

### Official Review · Reviewer_TD34 · 2023-11-01

**Soundness:** 2 fair
**Presentation:** 3 good
**Contribution:** 3 good
**Rating:** 5
**Confidence:** 4

**Summary:**

- This manuscript introduces a new hot-plugging adapter, with which the task-specific model's foundation backbone can be replaced without re-training for both the backbone and the task-specific head.
- The proposed method is tested on the CLIP series foundation models and evaluated on various vision-language tasks. The results validate the proposed method's effectiveness. Specifically, the performance of downstream tasks is improved when the backbone networks are replaced with more powerful ones.

**Strengths:**

1. The idea of hot-plugging adapters is interesting. It could have a good impact on future research and other applications.
2. The proposed method is technically sound.
3. The manuscript is well-written and easy to follow.
4. The comprehensive experiments validate the model's effectiveness on various tasks.

**Weaknesses:**

1. While the idea of hot-plugging adapters is intuitively sound at first glance, this paper lacks quantitative evidence to support this motivation. Specifically, the motivation of this paper is: *When replacing the visual backbones, fine-tuning the proposed adapter is better than fine-tuning the downstream task-specific head*. Therefore, a comparison between these two fine-tuning methods should be presented, and such a comparison should be in a fair enough setting because, in my opinion, it should be the most important experiment for the whole manuscript. Specifically, the author should compare 1) the trainable parameter amounts, 2) training FLOPs, 3) The data amounts needed for fine-tuning, and 4) the fine-tuning schedule of these two fine-tuning paradigms. In addition, the results in Table 1 show that the *TACA-BetterModel*'s performance is inferior to directly fine-tuning the task-specific head with a *BetterModel*, i.g., ViT-H, which also shows the necessity of such a comparison.
2. The symmetric adapter is a very interesting point of this manuscript, as it outperforms the standard adapter. It would be better to include some experiments to study its effectiveness on other tasks, e.g., image generation or some NLP tasks.
3. The manuscript should include more experiments to show its generalization ability to other non-CLIP models. For example, can the proposed method work on classic vision tasks, like detection or segmentation?
4. I am also curious if the method can be applied to replacing *different* foundation models. For example, can we use it to replace a DINO ViT with an ImageNet-22k Supervised ViT?
5. The proposed head-fixing strategy makes me think about the head-inheriting trick that is commonly used in knowledge distillation [a][b]. Therefore, discussing it in the related work section will increase the comprehensiveness of this work.

To conclude, I like the motivation of this work, and I also acknowledge that the provided experiments do validate its effectiveness to some extent. If the authors can address my concerns, especially the first point, I am happy to raise my rating.


[a] Yang, Chenhongyi, et al. "Prediction-guided distillation for dense object detection." European Conference on Computer Vision. Cham: Springer Nature Switzerland, 2022.

[b] Yang, Zhendong, et al. "Focal and global knowledge distillation for detectors." Proceedings of the IEEE/CVF Conference on Computer Vision and Pattern Recognition. 2022.

**Questions:**

N/A

---

> ### Author Response · Authors · 2023-11-22
>
> **Q1: About the motivation of hot-plugging adapters.**
>
> **A1:** (1) We acknowledge the confusion in our initial explanation. Our aim is to highlight **not** the superiority of adapter fine-tuning over task-specific head fine-tuning, but rather the **inefficiencies and inflexibility** of the latter. Hot-plugging adapters offer a **task-agnostic, generalized**, and **efficient** alternative, enhancing the performance of various downstream tasks without incurring training costs.
>
> (2) As shown in Table 1, direct fine-tuning of downstream heads can outperform TaCA, but it remains task-specific. The fine-tuning schedules and inference details for these paradigms are presented in Appendix C.  In the video-text retrieval task, we fine-tune three distinct sets of downstream heads on the MSRVTT, MSVD, and DiDeMo datasets, resulting in the training of a total of 42M parameters. For the video classification task, fine-tuning the classification head requires nearly twice as many parameters as the TaCA model. However, TaCA requires training just once, with no additional fine-tuning costs, yielding a more parameter-efficient and training-free application in downstream tasks. This discussion will be incorporated into the final paper.
>
>
>
> | **Strategy**   | **Learnable params.** | **Finetune data**      | **MSRVTT** | **MSVD** | **DiDeMo** |
> | -------------- | --------------------- | ---------------------- | ---------- | -------- | ---------- |
> | **Finetuning** | 14M                   | WIT+MSRVTT/MSVD/DiDeMo | 44.8       | 46.7     | 38.1       |
> | **TaCA**       | 17M                   | No need                | 44.5       | 45.6     | 36.6       |
>
> Table1: Comparison on Video-Text Retrieval datasets. The new visual fundation model (VFM) is ViT-L/14, and the old VFM is ViT-B/16.
>
> | **Strategy**   | **Learnable params.** | **Finetune data** | **K400** | **UCF-101** |
> | -------------- | --------------------- | ----------------- | -------- | ----------- |
> | **Finetuning** | 29M                   | Kinetics-400      | 87       | 85.7        |
> | **TaCA**       | 17M                   | No need           | 83.6     | 83.1        |
>
> Table2: Comparison on Video Classification datasets. The new visual fundation model (VFM) is ViT-L/14, and the old VFM is ViT-B/16.
>
>
>
> **Q2: Expanding Experiments to Additional Tasks (e.g., image generation or some NLP tasks).**
>
> **A2:** We appreciate your suggestion. Our extension to other downstream tasks, such as image generation, is documented in Appendix C.4, page 17. Although the styles of generated images are subtly differentiated, our method is adept at maintaining detailed semantics. This represents the early phase of our research; future work will aim at refining these solutions.
>
>
>
> **Q3: Generalization to Non-CLIP Models and Classic Vision Tasks. Can the proposed method work on classic vision tasks, like detection or segmentation?**
>
> **A3:** We have validated TaCA's effectiveness on non-CLIP models, such as Dinov2 and BEiT, in Table 9 (page 9). These models show slight improvements and meet the compatibility criterion for various tasks.
>
> We have also applied our method to classical vision tasks, such as zero-shot image classification and zero-shot image-text retrieval, with results presented in Tables 1 and 2 demonstrating consistent improvements. The potential application of TaCA to detection and segmentation will be explored in forthcoming studies.
>
>
>
>
>
> | **Model**         | **ImageNet** | **CIFAR100** |
> | ----------------- | ------------ | ------------ |
> | **ViT-B/16**      | 68.6         | 68.7         |
> | **ViT-L/14**      | 75.3         | 77.9         |
> | **TaCA-ViT-L/14** | 74.3(+5.7)   | 76.5(+7.8)   |
>
> Table1: Zero-shot image classification on ImageNet and CIFAR100 datasets. We report the accuracy.
>
> | **Model**         | **MSCOCO**   | **Flickr**  |
> | ----------------- | ------------ | ----------- |
> | **ViT-B/16**      | 42.75        | 72.18       |
> | **ViT-L/14**      | 46.42        | 75.08       |
> | **TaCA-ViT-L/14** | 44.71(+3.67) | 73.97(+2.9) |
>
> Table2: Zero-shot image-text retrieval on MSCOCO and Flickr datasets. We report the Recall@1.
>
>
>
> **Q4: Does TaCA work well when replacing *different* foundation models? Can we use it to replace a DINO ViT with an ImageNet-22k Supervised ViT?**
>
> **A4:** TaCA's adaptability across diverse foundational backbones is discussed in Table 9 of the main paper. While it is possible to replace a DINO ViT with an ImageNet-22k Supervised ViT using TaCA, the performance may be suboptimal if the new foundation model's representational capabilities are inferior. The efficacy of downstream tasks is contingent upon the foundational models' improved representational quality.
>
> **Q5: Incorporating Knowledge Distillation in Related Work**
>
> **A5:** We have included a new subsection on knowledge distillation in the related works of our revised manuscript, which can be found highlighted on page 18.

---

> > ### Author Response · Authors · 2023-11-23
> > **Looking forward to the further discussion**
> >
> > We greatly value your time and insightful feedbacks. In response to your suggestions,  we've included additional contents, that are:
> >
> > 1. Quantitative analysis on two different finetuning strategies.
> > 2. More experiments on downstream tasks, such as zero-shot image classification and zero-shot image-text retrieval.
> > 3. Additional related works in the revised manuscript.
> >
> > We eagerly anticipate your reconsideration and further discussion.

---

### Meta-Review · Area_Chair_7Sha · 2023-12-06

**Metareview:**

This paper proposes an adapter, which is used to task-agnostically adapt a model's representations such that it can be used when replacing the basemodel with another model without having to retraining task-specific head modules that were trained on the original model. To this end, a novel adapter is introduced and trained with within-modal and cross-modal distillation objective.
The proposed method is tested on the CLIP and other VLM models and evaluated on various vision-language tasks. The results validate the proposed method's effectiveness. Specifically, the performance of downstream tasks is improved when the backbone networks are replaced with more powerful ones.

**Justification For Why Not Higher Score:**

No results for language models and limited performance compared to re-finetuning on downstream tasks. The latter is crucial as ultimately not having the highest downstream performance presents a big issue for deployment and applications. Furthermore, while the number of retrained parameters does grow by the number of downstream tasks, heads are fairly cheap to finetune and approaches like LoRA further save large amounts of parameters to be saved.

**Justification For Why Not Lower Score:**

N/A

---

### Decision · Program_Chairs · 2024-01-16

Reject